# Batched Multi-armed Bandits Problem

**Zijun Gao, Yanjun Han, Zhimei Ren, Zhengqing Zhou**
Department of {Statistics, Electrical Engineering, Statistics, Mathematics}
Stanford University
{zijungao,yjhan,zren,zqzhou}@stanford.edu

## Abstract

In this paper, we study the multi-armed bandit problem in the batched setting where the employed policy must split data into a small number of batches. While the minimax regret for the two-armed stochastic bandits has been completely characterized in [PRCS16], the effect of the number of arms on the regret for the multi-armed case is still open. Moreover, the question whether adaptively chosen batch sizes will help to reduce the regret also remains underexplored. In this paper, we propose the BaSE (batched successive elimination) policy to achieve the rate-optimal regrets (within logarithmic factors) for batched multi-armed bandits, with matching lower bounds even if the batch sizes are determined in an adaptive manner.

## 1 Introduction and Main Results

Batch learning and online learning are two important aspects of machine learning, where the learner is a passive observer of a given collection of data in batch learning, while he can actively determine the data collection process in online learning. Recently, the combination of these learning procedures has been arised in an increasing number of applications, where the active querying of data is possible but limited to a fixed number of rounds of interaction. For example, in clinical trials [Tho33, Rob52], data come in batches where groups of patients are treated simultaneously to design the next trial. In crowdsourcing [KCS08], it takes the crowd some time to answer the current queries, so that the total time constraint imposes restrictions on the number of rounds of interaction. Similar problems also arise in marketing [BM07] and simulations [CG09].

In this paper we study the influence of round constraints on the learning performance via the following batched multi-armed bandit problem. Let $\mathcal{I} = \{1, 2, \cdots, K\}$ be a given set of $K \geq 2$ arms of a stochastic bandit, where successive pulls of an arm $i \in \mathcal{I}$ yields rewards which are i.i.d. samples from distribution $\nu^{(i)}$ with mean $\mu^{(i)}$. Throughout this paper we assume that the reward follows a Gaussian distribution, i.e., $\nu^{(i)} = \mathcal{N}(\mu^{(i)}, 1)$, where generalizations to general sub-Gaussian rewards and variances are straightforward. Let $\mu^\star = \max_{i \in [K]} \mu^{(i)}$ be the expected reward of the best arm, and $\Delta_i = \mu^\star - \mu^{(i)} \geq 0$ be the gap between arm $i$ and the best arm. The entire time horizon $T$ is splitted into $M$ batches represented by a *grid* $\mathcal{T} = \{t_1, \cdots, t_M\}$, with $1 \leq t_1 < t_2 < \cdots < t_M = T$, where the grid belongs to one of the following categories:

1. Static grid: the grid $\mathcal{T} = \{t_1, \cdots, t_M\}$ is fixed ahead of time, before sampling any arms;
2. Adaptive grid: for $j \in [M]$, the grid value $t_j$ may be determined after observing the rewards up to time $t_{j-1}$ and using some external randomness.

Note that the adaptive grid is more powerful and practical than the static one, and we recover batch learning and online learning by setting $M = 1$ and $M = T$, respectively. A sampling policy $\pi = (\pi_t)_{t=1}^T$ is a sequence of random variables $\pi_t \in [K]$ indicating which arm to pull at time $t \in [T]$, where for $t_{j-1} < t \leq t_j$, the policy $\pi_t$ depends only on observations up to time $t_{j-1}$. In other words,

the policy $\pi_t$ only depends on observations strictly anterior to the current batch of $t$. The ultimate goal is to devise a sampling policy $\pi$ to minimize the expected cumulative regret (or pseudo-regret, or simply *regret*), i.e., to minimize $\mathbb{E}[R_T(\pi)]$ where

$$R_T(\pi) \triangleq \sum_{t=1}^{T} \left( \mu^\star - \mu^{(\pi_t)} \right) = T\mu^\star - \sum_{t=1}^{T} \mu^{(\pi_t)}.$$

Let $\Pi_{M,T}$ be the set of policies with $M$ batches and time horizon $T$, our objective is to characterize the following *minimax regret* and *problem-dependent regret* under the batched setting:

$$R^\star_{\text{min-max}}(K, M, T) \triangleq \inf_{\pi \in \Pi_{M,T}} \sup_{\{\mu^{(i)}\}_{i=1}^{K} : \Delta_i \leq \sqrt{K}} \mathbb{E}[R_T(\pi)], \tag{1}$$

$$R^\star_{\text{pro-dep}}(K, M, T) \triangleq \inf_{\pi \in \Pi_{M,T}} \sup_{\Delta > 0} \Delta \cdot \sup_{\{\mu^{(i)}\}_{i=1}^{K} : \Delta_i \in \{0\} \cup [\Delta, \sqrt{K}]} \mathbb{E}[R_T(\pi)]. \tag{2}$$

Note that the gaps between arms can be arbitrary in the definition of the minimax regret, while a lower bound on the minimum gaps is present in the problem-dependent regret. The constraint $\Delta_i \leq \sqrt{K}$ is a technical condition in both scenarios, which is more relaxed than the usual condition $\Delta_i \in [0, 1]$. These quantities are motivated by the fact that, when $M = T$, the upper bounds of the regret for multi-armed bandits usually take the form [Vog60, LR85, AB09, BPR13, PR13]

$$\mathbb{E}[R_T(\pi^1)] \leq C\sqrt{KT},$$

$$\mathbb{E}[R_T(\pi^2)] \leq C \sum_{i \in [K] : \Delta_i > 0} \frac{\max\{1, \log(T\Delta_i^2)\}}{\Delta_i},$$

where $\pi^1, \pi^2$ are some policies, and $C > 0$ is an absolute constant. These bounds are also tight in the minimax sense [LR85, AB09]. As a result, in the fully adaptive setting (i.e., when $M = T$), we have the optimal regrets $R^\star_{\text{min-max}}(K, T, T) = \Theta(\sqrt{KT})$, and $R^\star_{\text{pro-dep}}(K, T, T) = \Theta(K \log T)$. The target is to find the dependence of these quantities on the number of batches $M$.

Our first result tackles the upper bounds on the minimax regret and problem-dependent regret.

**Theorem 1.** *For any $K \geq 2, T \geq 1, 1 \leq M \leq T$, there exist two policies $\pi^1$ and $\pi^2$ under static grids (explicitly defined in Section 2) such that if $\max_{i \in [K]} \Delta_i \leq \sqrt{K}$, we have*

$$\mathbb{E}[R_T(\pi^1)] \leq \mathsf{polylog}(K, T) \cdot \sqrt{K} T^{\frac{1}{2-2^{1-M}}},$$

$$\mathbb{E}[R_T(\pi^2)] \leq \mathsf{polylog}(K, T) \cdot \frac{K T^{1/M}}{\min_{i \neq \star} \Delta_i},$$

*where $\mathsf{polylog}(K, T)$ hides poly-logarithmic factors in $(K, T)$.*

The following corollary is immediate.

**Corollary 1.** *For the $M$-batched $K$-armed bandit problem with time horizon $T$, it is sufficient to have $M = O(\log \log T)$ batches to achieve the optimal minimax regret $\Theta(\sqrt{KT})$, and $M = O(\log T)$ to achieve the optimal problem-dependent regret $\Theta(K \log T)$, where both optimal regrets are within logarithmic factors.*

For the lower bounds of the regret, we treat the static grid and the adaptive grid separately. The next theorem presents the lower bounds under any static grid.

**Theorem 2.** *For the $M$-batched $K$-armed bandit problem with time horizon $T$ and any static grid, the minimax and problem-dependent regrets can be lower bounded as*

$$R^\star_{\text{min-max}}(K, M, T) \geq c \cdot \sqrt{K} T^{\frac{1}{2-2^{1-M}}},$$

$$R^\star_{\text{pro-dep}}(K, M, T) \geq c \cdot K T^{\frac{1}{M}},$$

*where $c > 0$ is a numerical constant independent of $K, M, T$.*

We observe that for any static grids, the lower bounds in Theorem 2 match those in Theorem 1 within poly-logarithmic factors. For general adaptive grids, the following theorem shows regret lower bounds which are slightly weaker than Theorem 2.

**Theorem 3.** *For the $M$-batched $K$-armed bandit problem with time horizon $T$ and any adaptive grid, the minimax and problem-dependent regrets can be lower bounded as*

$$R_{\text{min-max}}^\star(K, M, T) \geq cM^{-2} \cdot \sqrt{K}T^{\frac{1}{2-2^{1-M}}},$$

$$R_{\text{pro-dep}}^\star(K, M, T) \geq cM^{-2} \cdot KT^{\frac{1}{M}},$$

*where $c > 0$ is a numerical constant independent of $K, M, T$.*

Compared with Theorem 2, the lower bounds in Theorem 3 lose a polynomial factor in $M$ due to a larger policy space. However, since the number of batches $M$ of interest is at most $O(\log T)$ (otherwise by Corollary 1 we effectively arrive at the fully adaptive case with $M = T$), this penalty is at most poly-logarithmic in $T$. Consequently, Theorem 3 shows that for any adaptive grid, albeit conceptually more powerful, its performance is essentially no better than that of the best static grid. Specifically, we have the following corollary.

**Corollary 2.** *For the $M$-batched $K$-armed bandit problem with time horizon $T$ with either static or adaptive grids, it is necessary to have $M = \Omega(\log \log T)$ batches to achieve the optimal minimax regret $\Theta(\sqrt{KT})$, and $M = \Omega(\log T / \log \log T)$ to achieve the optimal problem-dependent regret $\Theta(K \log T)$, where both optimal regrets are within logarithmic factors.*

In summary, the above results have completely characterized the minimax and problem-dependent regrets for batched multi-armed bandit problems, within logarithmic factors. It is an outstanding open question whether the $M^{-2}$ term in Theorem 3 can be removed using more refined arguments.

## 1.1 Related works

The multi-armed bandits problem is an important class of sequential optimization problems which has been extensively studied in various fields such as statistics, operations research, engineering, computer science and economics over the recent years [BCB12]. In the fully adaptive scenario, the regret analysis for stochastic bandits can be found in [Vog60, LR85, BK97, ACBF02, AB09, AMS09, AB10, AO10, GC11, BPR13, PR13].

There is less attention on the batched setting with limited rounds of interaction. The batched setting is studied in [CBDS13] under the name of switching costs, where it is shown that $O(\log \log T)$ batches are sufficient to achieve the optimal minimax regret. For small number of batches $M$, the batched two-armed bandit problem is studied in [PRCS16], where the results of Theorems 1 and 2 are obtained for $K = 2$. However, the generalization to the multi-armed case is not straightforward, and more importantly, the practical scenario where the grid is adaptively chosen based on the historic data is excluded in [PRCS16]. For the multi-armed case, a different problem of finding the best $k$ arms in the batched setting has been studied in [JJNZ16, AAAK17], where the goal is pure exploration, and the error dependence on the time horizon decays super-polynomially. We also refer to [DRY18] for a similar setting with convex bandits and best arm identification. The regret analysis for batched stochastic multi-armed bandits still remains underexplored.

We also review some literature on general computation with limited rounds of adaptivity, and in particular, on the analysis of lower bounds. In theoretical computer science, this problem has been studied under the name of parallel algorithms for certain tasks (e.g., sorting and selection) given either deterministic [Val75, BT83, AA88] or noisy outcomes [FRPU94, DKMR14, BMW16]. In (stochastic) convex optimization, the information-theoretic limits are typically derived under the oracle model where the oracle can be queried adaptively [NY83, AWBR09, Sha13, DRY18]. However, in the previous works, one usually optimizes the sampling distribution over a fixed sample size at each step, while it is more challenging to prove lower bounds for policies which can also determine the sample size. There is one exception [AAAK17], whose proof relies on a complicated decomposition of near-uniform distributions. Hence, our technique of proving Theorem 3 is also expected to be an addition to these literatures.

## 1.2 Organization

The rest of this paper is organized as follows. In Section 2, we introduce the BaSE policy for general batched multi-armed bandit problems, and show that it attains the upper bounds in Theorem 1 under two specific grids. Section 3 presents the proofs of lower bounds for both the minimax and

problem-dependent regrets, where Section 3.1 deals with the static grids and Section 3.2 tackles the adaptive grids. Experimental results are presented in Section 4. The auxiliary lemmas and the proof of main lemmas are deferred to supplementary materials.

## 1.3 Notations

For a positive integer $n$, let $[n] \triangleq \{1, \cdots, n\}$. For any finite set $A$, let $|A|$ be its cardinality. We adopt the standard asymptotic notations: for two non-negative sequences $\{a_n\}$ and $\{b_n\}$, let $a_n = O(b_n)$ iff $\limsup_{n \to \infty} a_n/b_n < \infty$, $a_n = \Omega(b_n)$ iff $b_n = O(a_n)$, and $a_n = \Theta(b_n)$ iff $a_n = O(b_n)$ and $b_n = O(a_n)$. For probability measures $P$ and $Q$, let $P \otimes Q$ be the product measure with marginals $P$ and $Q$. If measures $P$ and $Q$ are defined on the same probability space, we denote by $\mathsf{TV}(P, Q) = \frac{1}{2} \int |dP - dQ|$ and $D_{\mathrm{KL}}(P\|Q) = \int dP \log \frac{dP}{dQ}$ the total variation distance and Kullback–Leibler (KL) divergences between $P$ and $Q$, respectively.

# 2 The BaSE Policy

In this section, we propose the BaSE policy for the batched multi-armed bandit problem based on successive elimination, as well as two choices of the static grids to prove Theorem 1.

## 2.1 Description of the policy

The policy that achieves the optimal regrets is essentially adapted from Successive Elimination (SE). The original version of SE was introduced in [EDMM06], and [PR13] shows that in the $M = T$ case SE achieves both the optimal minimax and problem-dependent rates. Here we introduce a batched version of SE called Batched Successive Elimination (BaSE) to handle the general case $M \leq T$.

Given a pre-specified grid $\mathcal{T} = \{t_1, \cdots, t_M\}$, the idea of the BaSE policy is simply to explore in the first $M - 1$ batches and then commit to the best arm in the last batch. At the end of each exploration batch, we remove arms which are probably bad based on past observations. Specfically, let $\mathcal{A} \subseteq \mathcal{I}$ denote the *active* arms that are candidates for the optimal arm, where we initialize $\mathcal{A} = \mathcal{I}$ and sequentially drop the arms which are "significantly" worse than the "best" one. For the first $M - 1$ batches, we pull all active arms for a same number of times (neglecting rounding issues[1]) and eliminate some arms from $\mathcal{A}$ at the end of each batch. For the last batch, we commit to the arm in $\mathcal{A}$ with maximum average reward.

Before stating the exact algorithm, we introduce some notations. Let

$$\bar{Y}^i(t) = \frac{1}{|\{s \leq t : \text{arm } i \text{ is pulled at time } s\}|} \sum_{s=1}^{t} Y_s \mathbb{1}\{\text{arm } i \text{ is pulled at time } s\}$$

denote the average rewards of the arm $i$ up to time $t$, and $\gamma > 0$ is a tuning parameter associated with the UCB bound. The algorithm is described in detail in Algorithm 1.

Note that the BaSE algorithm is not fully specified unless the grid $\mathcal{T}$ is determined. Here we provide two choices of static grids which are similar to [PRCS16] as follows: let

$$u_1 = a, \quad u_m = a\sqrt{u_{m-1}}, \quad m = 2, \cdots, M, \qquad t_m = \lfloor u_m \rfloor, \quad m \in [M],$$
$$u_1' = b, \quad u_m' = bu_{m-1}', \quad m = 2, \cdots, M, \qquad t_m' = \lfloor u_m' \rfloor, \quad m \in [M],$$

where parameters $a, b$ are chosen appropriately such that $t_M = t_M' = T$, i.e.,

$$a = \Theta\left(T^{\frac{1}{2-2^{1-M}}}\right), \qquad b = \Theta\left(T^{\frac{1}{M}}\right). \tag{3}$$

For minimizing the minimax regret, we use the "minimax" grid defined by $\mathcal{T}_{\mathrm{minimax}} = \{t_1, \cdots, t_M\}$; as for the problem-dependent regret, we use the "geometric" grid which is defined by $\mathcal{T}_{\mathrm{geometric}} = \{t_1', \cdots, t_M'\}$. We will denote by $\pi_{\mathrm{BaSE}}^1$ and $\pi_{\mathrm{BaSE}}^2$ the respective policies under these grids.

**Algorithm 1:** Batched Successive Elimination (BaSE)

---

**Input:** Arms $\mathcal{I} = [K]$; time horizon $T$; number of batches $M$; grid $\mathcal{T} = \{t_1, ..., t_M\}$; tuning parameter $\gamma$.
**Initialization:** $\mathcal{A} \leftarrow \mathcal{I}$.
**for** $m \leftarrow 1$ **to** $M - 1$ **do**

    (a) During the period $[t_{m-1} + 1, t_m]$, pull an arm from $\mathcal{A}$ for a same number of times.
    (b) At time $t_m$:
    Let $\bar{Y}^{\max}(t_m) = \max_{j \in \mathcal{A}} \bar{Y}^j(t_m)$, and $\tau_m$ be the total number of pulls of each arm in $\mathcal{A}$.
    **for** $i \in \mathcal{A}$ **do**
        **if** $\bar{Y}^{\max}(t_m) - \bar{Y}^i(t_m) \geq \sqrt{\gamma \log(TK)/\tau_m}$ **then**
            $\mathcal{A} \leftarrow \mathcal{A} - \{i\}$.
        **end**
    **end**
**end**
**for** $t \leftarrow t_{M-1} + 1$ **to** $T$ **do**
    pull arm $i_0$ such that $i_0 \in \arg\max_{j \in \mathcal{A}} \bar{Y}^j(t_{M-1})$ (break ties arbitrarily).
**end**
**Output: Resulting policy** $\pi$.

---

## 2.2 Regret analysis

The performance of the BaSE policy is summarized in the following theorem.

**Theorem 4.** *Consider an $M$-batched, $K$-armed bandit problem where the time horizon is $T$. let $\pi^1_{\text{BaSE}}$ be the* BaSE *policy equipped with the grid $\mathcal{T}_{\text{minimax}}$ and $\pi^2_{\text{BaSE}}$ be the* BaSE *policy equipped with the grid $\mathcal{T}_{\text{geometric}}$. For $\gamma \geq 12$ and $\max_{i \in [K]} \Delta_i = O(\sqrt{K})$, we have*

$$\mathbb{E}[R_T(\pi^1_{\text{BaSE}})] \leq C \log K \sqrt{\log(KT)} \cdot \sqrt{K} T^{\frac{1}{2 - 2^{1-M}}}, \tag{4}$$

$$\mathbb{E}[R_T(\pi^2_{\text{BaSE}})] \leq C \log K \log(KT) \cdot \frac{KT^{1/M}}{\min_{i \neq \star} \Delta_i}, \tag{5}$$

*where $C > 0$ is a numerical constant independent of $K, M$ and $T$.*

Note that Theorem 4 implies Theorem 1. In the sequel we sketch the proof of Theorem 4, where the main technical difficulty is to appropriately control the number of pulls for each arm under batch constraints, where there is a random number of active arms in $\mathcal{A}$ starting from the second batch. We also refer to a recent work [EKMM19] for a tighter bound on the problem-dependent regret with an adaptive grid.

*Proof of Theorem 4.* For notational simplicity we assume that there are $K + 1$ arms, where arm $0$ is the arm with highest expected reward (denoted as $\star$), and $\Delta_i = \mu_\star - \mu_i \geq 0$ for $i \in [K]$. Define the following events: for $i \in [K]$, let $A_i$ be the event that arm $i$ is eliminated before time $t_{m_i}$, where

$$m_i = \min \left\{ j \in [M] : \text{arm } i \text{ has been pulled at least } \tau_i^\star \triangleq \frac{4\gamma \log(TK)}{\Delta_i^2} \text{ times before time } t_j \in \mathcal{T} \right\},$$

with the understanding that if the minimum does not exist, we set $m_i = M$ and the event $A_i$ occurs. Let $B$ be the event that arm $\star$ is not eliminated throughout the time horizon $T$. The final "good event" $E$ is defined as $E = (\cap_{i=1}^K A_i) \cap B$. We remark that $m_i$ is a random variable depending on the order in which the arms are eliminated. The following lemma shows that by our choice of $\gamma \geq 12$, the good event $E$ occurs with high probability.

**Lemma 1.** *The event $E$ happens with probability at least $1 - \frac{2}{TK}$.*

The proof of Lemma 1 is postponed to the supplementary materials. By Lemma 1, the expected regret $R_T(\pi)$ (with $\pi = \pi^1_{\text{BaSE}}$ or $\pi^2_{\text{BaSE}}$) when the event $E$ does not occur is at most

$$\mathbb{E}[R_T(\pi) \mathbb{1}(E^c)] \leq T \max_{i \in [K]} \Delta_i \cdot \mathbb{P}(E^c) = O(1). \tag{6}$$

Next we condition on the event $E$ and upper bound the regret $\mathbb{E}[R_T(\pi^1_{\mathrm{BaSE}})\mathbb{1}(E)]$ for the minimax grid $\mathcal{T}_{\mathrm{minimax}}$. The analysis of the geometric grid $\mathcal{T}_{\mathrm{geometric}}$ is entirely analogous, and is deferred to the supplementary materials.

For the policy $\pi^1_{\mathrm{BaSE}}$, let $\mathcal{I}_0 \subseteq \mathcal{I}$ be the (random) set of arms which are eliminated at the end of the first batch, $\mathcal{I}_1 \subseteq \mathcal{I}$ be the (random) set of remaining arms which are eliminated before the last batch, and $\mathcal{I}_2 = \mathcal{I} - \mathcal{I}_0 - \mathcal{I}_1$ be the (random) set of arms which remain in the last batch. It is clear that the total regret incurred by arms in $\mathcal{I}_0$ is at most $t_1 \cdot \max_{i\in[K]} \Delta_i = O(\sqrt{K}a)$, and it remains to deal with the sets $\mathcal{I}_1$ and $\mathcal{I}_2$ separately.

For arm $i \in \mathcal{I}_1$, let $\sigma_i$ be the (random) number of arms which are eliminated *before* the arm $i$. Observe that the fraction of pullings of arm $i$ is at most $\frac{1}{K-\sigma_i}$ before arm $i$ is eliminated. Moreover, by the definition of $t_{m_i}$, we must have

$$\tau_i^\star > (\text{number of pullings of arm } i \text{ before } t_{m_i-1}) \geq \frac{t_{m_i-1}}{K} \implies \Delta_i\sqrt{t_{m_i-1}} \leq \sqrt{4\gamma K\log(TK)}.$$

Hence, the total regret incurred by pulling an arm $i \in \mathcal{I}_1$ is at most (note that $t_j \leq 2a\sqrt{t_{j-1}}$ for any $j = 2,3,\cdots,M$ by the choice of the grid)

$$\Delta_i \cdot \frac{t_{m_i}}{K-\sigma_i} \leq \Delta_i \cdot \frac{2a\sqrt{t_{m_i-1}}}{K-\sigma_i} \leq \frac{2a\sqrt{4\gamma K\log(TK)}}{K-\sigma_i}.$$

Note that there are at most $t$ elements in $(\sigma_i : i \in \mathcal{I}_1)$ which are at least $K - t$ for any $t = 2,\cdots,K$, the total regret incurred by pulling arms in $\mathcal{I}_1$ is at most

$$\sum_{i\in\mathcal{I}_1} \frac{2a\sqrt{4\gamma K\log(TK)}}{K-\sigma_i} \leq 2a\sqrt{4\gamma K\log(TK)} \cdot \sum_{t=2}^{K} \frac{1}{t} \leq 2a\log K\sqrt{4\gamma K\log(TK)}. \quad (7)$$

For any arm $i \in \mathcal{I}_2$, by the previous analysis we know that $\Delta_i\sqrt{t_{M-1}} \leq \sqrt{4\gamma K\log(TK)}$. Hence, let $T_i$ be the number of pullings of arm $i$, the total regret incurred by pulling arm $i \in \mathcal{I}_2$ is at most

$$\Delta_i T_i \leq T_i\sqrt{\frac{4\gamma K\log(TK)}{t_{M-1}}} \leq \frac{T_i}{T} \cdot 2a\sqrt{4\gamma K\log(TK)},$$

where in the last step we have used that $T = t_M \leq 2a\sqrt{t_{M-1}}$ in the minimax grid $\mathcal{T}_{\mathrm{minimax}}$. Since $\sum_{i\in\mathcal{I}_2} T_i \leq T$, the total regret incurred by pulling arms in $\mathcal{I}_2$ is at most

$$\sum_{i\in\mathcal{I}_2} \frac{T_i}{T} \cdot 2a\sqrt{4\gamma K\log(TK)} \leq 2a\sqrt{4\gamma K\log(TK)}. \quad (8)$$

By (7) and (8), the inequality

$$R_T(\pi^1_{\mathrm{BaSE}})\mathbb{1}(E) \leq 2a\sqrt{4\gamma K\log(TK)}(\log K + 1) + O(\sqrt{K}a)$$

holds almost surely. Hence, this inequality combined with (6) and the choice of $a$ in (3) yields the desired upper bound (4). $\qquad\square$

## 3 Lower Bound

This section presents lower bounds for the batched multi-armed bandit problem, where in Section 3.1 we design a fixed multiple hypothesis testing problem to show the lower bound for any policies under static grids, while in Section 3.2 we construct different hypotheses for different policies under general adaptive grids.

### 3.1 Static grid

The proof of Theorem 2 relies on the following lemma.

**Lemma 2.** *For any static grid $0 = t_0 < t_1 < \cdots < t_M = T$ and the smallest gap $\Delta \in (0, \sqrt{K}]$, the following minimax lower bound holds for any policy $\pi$ under this grid:*

$$\sup_{\{\mu^{(i)}\}_{i=1}^K : \Delta_i \in \{0\} \cup [\Delta, \sqrt{K}]} \mathbb{E}[R_T(\pi)] \geq \Delta \cdot \sum_{j=1}^M \frac{t_j - t_{j-1}}{4} \exp\left(-\frac{2t_{j-1}\Delta^2}{K-1}\right).$$

We first show that Lemma 2 implies Theorem 2 by choosing the smallest gap $\Delta > 0$ appropriately. By definitions of the minimax regret $R_{\text{min-max}}^\star$ and the problem-dependent regret $R_{\text{pro-dep}}^\star$, choosing $\Delta = \Delta_j = \sqrt{(K-1)/(t_{j-1}+1)} \in [0, \sqrt{K}]$ in Lemma 2 yields that

$$R_{\text{min-max}}^\star(K, M, T) \geq c_0 \sqrt{K} \cdot \max_{j \in [M]} \frac{t_j}{\sqrt{t_{j-1}+1}},$$

$$R_{\text{pro-dep}}^\star(K, M, T) \geq c_0 K \cdot \max_{j \in [M]} \frac{t_j}{t_{j-1}+1},$$

for some numerical constant $c_0 > 0$. Since $t_0 = 0, t_M = T$, the lower bounds in Theorem 2 follow.

Next we employ the general idea of the multiple hypothesis testing to prove Lemma 2. Consider the following $K$ candidate reward distributions:

$$P_1 = \mathcal{N}(\Delta, 1) \otimes \mathcal{N}(0, 1) \otimes \mathcal{N}(0, 1) \otimes \cdots \otimes \mathcal{N}(0, 1),$$
$$P_2 = \mathcal{N}(\Delta, 1) \otimes \mathcal{N}(2\Delta, 1) \otimes \mathcal{N}(0, 1) \otimes \cdots \otimes \mathcal{N}(0, 1),$$
$$P_3 = \mathcal{N}(\Delta, 1) \otimes \mathcal{N}(0, 1) \otimes \mathcal{N}(2\Delta, 1) \otimes \cdots \otimes \mathcal{N}(0, 1),$$
$$\vdots$$
$$P_K = \mathcal{N}(\Delta, 1) \otimes \mathcal{N}(0, 1) \otimes \mathcal{N}(0, 1) \otimes \cdots \otimes \mathcal{N}(2\Delta, 1).$$

We remark that this construction is not entirely symmetric, where the reward distribution of the first arm is always $\mathcal{N}(\Delta, 1)$. The key properties of this construction are as follows:

1. For any $i \in [K]$, arm $i$ is the optimal arm under reward distribution $P_i$;
2. For any $i \in [K]$, pulling a wrong arm incurs a regret at least $\Delta$ under reward distribution $P_i$.

As a result, since the average regret serves as a lower bound of the worst-case regret, we have

$$\sup_{\{\mu^{(i)}\}_{i=1}^K : \Delta_i \in \{0\} \cup [\Delta, \sqrt{K}]} \mathbb{E}R_T(\pi) \geq \frac{1}{K} \sum_{i=1}^K \sum_{t=1}^T \mathbb{E}_{P_i^t} R^t(\pi) \geq \Delta \sum_{t=1}^T \frac{1}{K} \sum_{i=1}^K P_i^t(\pi_t \neq i), \quad (9)$$

where $P_i^t$ denotes the distribution of observations available at time $t$ under $P_i$, and $R^t(\pi)$ denotes the instantaneous regret incurred by the policy $\pi_t$ at time $t$. Hence, it remains to lower bound the quantity $\frac{1}{K} \sum_{i=1}^K P_i^t(\pi_t \neq i)$ for any $t \in [T]$, which is the subject of the following lemma.

**Lemma 3.** *Let $Q_1, \cdots, Q_n$ be probability measures on some common probability space $(\Omega, \mathcal{F})$, and $\Psi : \Omega \to [n]$ be any measurable function (i.e., test). Then for any tree $T = ([n], E)$ with vertex set $[n]$ and edge set $E$, we have*

$$\frac{1}{n} \sum_{i=1}^n Q_i(\Psi \neq i) \geq \sum_{(i,j) \in E} \frac{1}{2n} \exp(-D_{\text{KL}}(Q_i \| Q_j)).$$

The proof of Lemma 3 is deferred to the supplementary materials, and we make some remarks below.

**Remark 1.** *A more well-known lower bound for $\frac{1}{n} \sum_{i=1}^n Q_i(\Psi \neq i)$ is the Fano's inequality [CT06], which involves the mutual information $I(U; X)$ with $U \sim \text{Uniform}([n])$ and $P_{X|U=i} = Q_i$. However, since $I(U; X) = \mathbb{E}_{P_U} D_{\text{KL}}(P_{X|U} \| P_X)$, Fano's inequality gives a lower bound which depends linearly on the pairwise KL divergence rather than exponentially and is thus loose for our purpose.*

**Remark 2.** *An alternative lower bound is to use $\frac{1}{2n^2} \sum_{i \neq j} \exp(-D_{\text{KL}}(Q_i \| Q_j))$, i.e., the summation is taken over all pairs $(i, j)$ instead of just the edges in a tree. However, this bound is weaker than Lemma 3, and in the case where $Q_i = \mathcal{N}(i\Delta, 1)$ for some large $\Delta > 0$, Lemma 3 with the tree $T = ([n], \{(1, 2), (2, 3), \cdots, (n-1, n)\})$ is tight (giving the rate $(\exp(-O(\Delta^2)))$ while the alternative bound loses a factor of $n$ (giving the rate $\exp(-O(\Delta^2))/n$).*

To lower bound (9), we apply Lemma 3 with the star tree $T = ([n], \{(1,i) : 2 \le i \le n\})$. For $i \in [K]$, denote by $T_i(t)$ the number of pulls of arm $i$ anterior to the current batch of $t$. Hence, $\sum_{i=1}^{K} T_i(t) = t_{j-1}$ if $t \in (t_{j-1}, t_j]$. Moreover, since $D_{\text{KL}}(P_1^t \| P_i^t) = 2\Delta^2 \mathbb{E}_{P_1^t} T_i(t)$, we have

$$
\frac{1}{K} \sum_{i=1}^{K} P_i^t(\pi_t \ne i) \ge \frac{1}{2K} \sum_{i=2}^{K} \exp(-D_{\text{KL}}(P_1^t \| P_i^t)) = \frac{1}{2K} \sum_{i=2}^{K} \exp(-2\Delta^2 \mathbb{E}_{P_1^t} T_i(t))
$$

$$
\ge \frac{K-1}{2K} \exp\left( -\frac{2\Delta^2}{K-1} \mathbb{E}_{P_1^t} \sum_{i=2}^{K} T_i(t) \right) \ge \frac{1}{4} \exp\left( -\frac{2\Delta^2 t_{j-1}}{K-1} \right). \tag{10}
$$

Now combining (9) and (10) completes the proof of Lemma 2.

## 3.2 Adaptive grid

Now we investigate the case where the grid may be randomized, and be generated sequentially in an adaptive manner. Recall that in the previous section, we construct multiple fixed hypotheses and show that no policy under a static grid can achieve a uniformly small regret under all hypotheses. However, this argument breaks down even if the grid is only randomized but *not* adaptive, due to the non-convex (in $(t_1, \cdots, t_M)$) nature of the lower bound in Lemma 2. In other words, we might not hope for a single fixed multiple hypothesis testing problem to work for *all* policies. To overcome this difficulty, a subroutine in the proof of Theorem 3 is to construct appropriate hypotheses *after* the policy is given (cf. the proof of Lemma 4). We sketch the proof below.

We shall only prove the lower bound for the minimax regret, where the analysis of the problem-dependent regret is entirely analogous. Consider the following time $T_1, \cdots, T_M \in [1, T]$ and gaps $\Delta_1, \cdots, \Delta_M \in (0, \sqrt{K})$ with

$$
T_j = \lfloor T^{\frac{1-2^{-j}}{1-2^{-M}}} \rfloor, \qquad \Delta_j = \frac{\sqrt{K}}{36M} \cdot T^{-\frac{1-2^{1-j}}{2(1-2^{-M})}}, \qquad j \in [M]. \tag{11}
$$

Let $\mathcal{T} = \{t_1, \cdots, t_M\}$ be any adaptive grid, and $\pi$ be any policy under the grid $\mathcal{T}$. For each $j \in [M]$, we define the event $A_j = \{t_{j-1} < T_{j-1}, t_j \ge T_j\}$ under policy $\pi$ with the convention that $t_0 = 0, t_M = T$. Note that the events $A_1, \cdots, A_M$ form a partition of the entire probability space. We also define the following family of reward distributions: for $j \in [M-1], k \in [K-1]$ let

$$
P_{j,k} = \mathcal{N}(0,1) \otimes \cdots \otimes \mathcal{N}(0,1) \otimes \mathcal{N}(\Delta_j + \Delta_M, 1) \otimes \mathcal{N}(0,1) \otimes \cdots \otimes \mathcal{N}(0,1) \otimes \mathcal{N}(\Delta_M, 1),
$$

where the $k$-th component of $P_{j,k}$ has a non-zero mean. For $j = M$, we define

$$
P_M = \mathcal{N}(0,1) \otimes \cdots \otimes \mathcal{N}(0,1) \otimes \mathcal{N}(\Delta_M, 1).
$$

Note that this construction ensures that $P_{j,k}$ and $P_M$ only differs in the $k$-th component, which is crucial for the indistinguishability results in Lemma 5.

We will be interested in the following quantities:

$$
p_j = \frac{1}{K-1} \sum_{k=1}^{K-1} P_{j,k}(A_j), \quad j \in [M-1], \qquad p_M = P_M(A_M),
$$

where $P_{j,k}(A)$ denotes the probability of the event $A$ given the true reward distribution $P_{j,k}$ and the policy $\pi$. The importance of these quantities lies in the following lemmas.

**Lemma 4.** *If $p_j \ge \frac{1}{2M}$ for some $j \in [M]$, then we have*

$$
\sup_{\{\mu^{(i)}\}_{i=1}^{K} : \Delta_i \le \sqrt{K}} \mathbb{E}[R_T(\pi)] \ge cM^{-2} \cdot \sqrt{K} T^{\frac{1}{2-2^{1-M}}},
$$

*where $c > 0$ is a numerical constant independent of $(K, M, T)$ and $(\pi, \mathcal{T})$.*

**Lemma 5.** *The following inequality holds: $\sum_{j=1}^{M} p_j \ge \frac{1}{2}$.*

The detailed proofs of Lemma 4 and Lemma 5 are deferred to the supplementary materials, and we only sketch the ideas here. Lemma 4 states that, if any of the events $A_j$ occurs with a non-small

probability in the respective $j$-th *world* (i.e., under the mixture of $(P_{j,k} : k \in [K-1])$ or $P_M$), then the policy $\pi$ has a large regret in the worst case. The intuition behind Lemma 4 is that, if the event $t_{j-1} \leq T_{j-1}$ occurs under the reward distribution $P_{j,k}$, then the observations in the first $(j-1)$ batches are not sufficient to distinguish $P_{j,k}$ from its (carefully designed) perturbed version with size of perturbation $\Delta_j$. Furthermore, if in addition $t_j \geq T_j$ holds, then the total regret is at least $\Omega(T_j \Delta_j)$ due to the indistinguishability of the $\Delta_j$ perturbations in the first $j$ batches. Hence, if $A_j$ occurs with a fairly large probability, the resulting total regret will be large as well.

Lemma 5 complements Lemma 4 by stating that at least one $p_j$ should be large. Note that if all $p_j$ were defined in the same world, the partition structure of $A_1, \cdots, A_M$ would imply $\sum_{j \in [M]} p_j \geq 1$. Since the occurrence of $A_j$ cannot really help to distinguish the $j$-th world with later ones, Lemma 5 shows that we may still operate in the same world and arrive at a slightly smaller constant than 1.

Finally we show how Lemma 4 and Lemma 5 imply Theorem 3. In fact, by Lemma 5, there exists some $j \in [M]$ such that $p_j \geq (2M)^{-1}$. Then by Lemma 4 and the arbitrariness of $\pi$, we arrive at the desired lower bound in Theorem 3.

## 4  Experiments

This section contains some experimental results on the performances of BaSE policy under different grids. The default parameters are $T = 5 \times 10^4, K = 3, M = 3$ and $\gamma = 1$, and the mean reward is $\mu^\star = 0.6$ for the optimal arm and is $\mu = 0.5$ for all other arms. In addition to the minimax and geometric grids, we also experiment on the arithmetic grid with $t_j = jT/M$ for $j \in [M]$. Figure 1 (a)-(c) display the empirical dependence of the average BaSE regrets under different grids, together with the comparison with the centralized UCB1 algorithm [ACBF02] without any batch constraints. We observe that the minimax grid typically results in a smallest regret among all grids, and $M = 4$ batches appear to be sufficient for the BaSE performance to approach the centralized performance. We also compare our BaSE algorithm with the ETC algorithm in [PRCS16] for the two-arm case, and Figure 1 (d) shows that BaSE achieves lower regrets than ETC. The source codes of the experiment can be found in `https://github.com/Mathegineer/batched-bandit`.

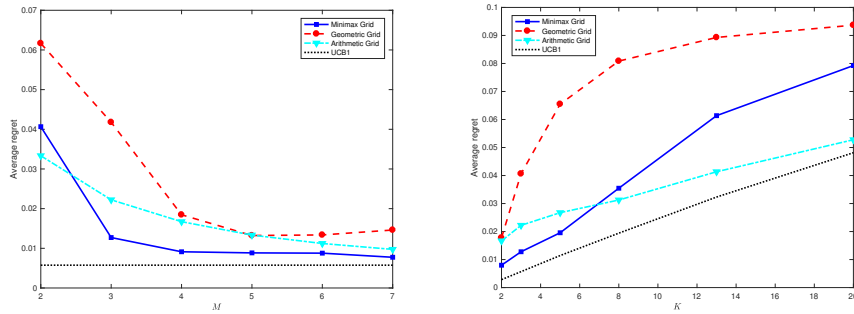

(a) Average regret vs. the number of batches $M$.   (b) Average regret vs. the number of arms $K$.

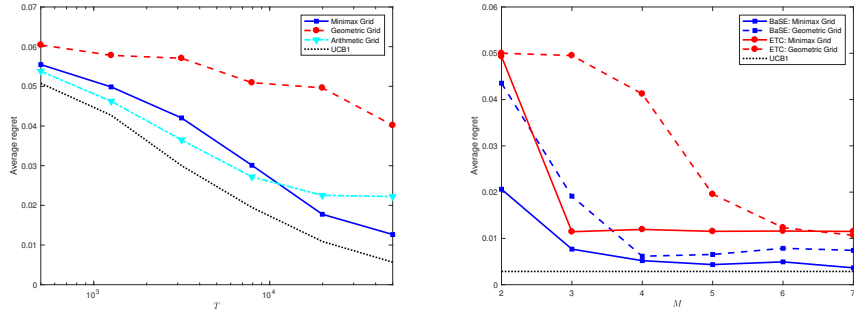

(c) Average regret vs. the time horizon $T$.          (d) Comparison of BaSE and ETC.

Figure 1: Empirical regret performances of the BaSE policy.

## Footnotes

[1]There might be some rounding issues here, and some arms may be pulled once more than others. In this case, the additional pull will not be counted towards the computation of the average reward $\bar{Y}^i(t)$, which ensures that all active arms are evaluated using the same number of pulls at the end of any batch. Note that in this way, the number of pulls for each arm is underestimated by at most half, therefore the regret analysis in Theorem 4 will give the same rate in the presence of rounding issues.

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
