[Supplementary Material]

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

# A  Auxiliary Lemmas

The following lemma is a generalization of [Tsy08, Lemma 2.6].

**Lemma 6.** *Let $P$ and $Q$ be any probability measures on the same probability space. Then*

$$\mathsf{TV}(P,Q) \leq \sqrt{1 - \exp(-D_{\mathsf{KL}}(P\|Q))} \leq 1 - \frac{1}{2}\exp\left(-D_{\mathsf{KL}}(P\|Q)\right).$$

*Proof.* Observe that the proof of [Tsy08, Lemma 2.6] gives

$$\left(\int \min\{dP, dQ\}\right)\left(\int \max\{dP, dQ\}\right) \geq \exp\left(-D_{\mathsf{KL}}(P\|Q)\right).$$

Since

$$\int \min\{dP, dQ\} = 1 - \mathsf{TV}(P,Q),$$

$$\int \max\{dP, dQ\} = 1 + \mathsf{TV}(P,Q),$$

the first inequality follows. The second inequality follows from the basic inequality $\sqrt{1-x} \leq 1-x/2$ for any $x \in [0,1]$. $\qquad\square$

The following lemma presents a graph-theoretic inequality, which is the crux of Lemma 3.

**Lemma 7.** *Let $T = (V, E)$ be a tree on $V = [n]$, and $x \in \mathbb{R}^n$ be any vector. Then*

$$\sum_{i=1}^{n} x_i - \max_{i \in [n]} x_i \geq \sum_{(i,j) \in E} \min\{x_i, x_j\}.$$

*Proof.* Without loss of generality we assume that $x_1 \leq x_2 \leq \cdots \leq x_n$. For any $k \in [n-1]$, we have

$$\sum_{(i,j) \in E} \mathbb{1}(\min\{x_i, x_j\} \geq x_k) = |\{(i,j) \in E : i \geq k, j \geq k\}| \leq n - k,$$

where the last inequality is due to the fact that restricting the tree $T$ on the vertices $\{k, k+1, \cdots, n\}$ is still acyclic. Hence,

$$
\begin{aligned}
\sum_{i=1}^{n} x_i - \max_{i \in [n]} x_i = \sum_{i=1}^{n-1} x_i &= (n-1)x_1 + \sum_{k=2}^{n-1}(n-k)(x_k - x_{k-1}) \\
&\geq (n-1)x_1 + \sum_{k=2}^{n-1}(x_k - x_{k-1}) \sum_{(i,j) \in E} \mathbb{1}(\min\{x_i, x_j\} \geq x_k) \\
&= \sum_{(i,j) \in E}\left(x_1 + \sum_{k=2}^{n-1}(x_k - x_{k-1})\mathbb{1}(\min\{x_i, x_j\} \geq x_k)\right) \\
&= \sum_{(i,j) \in E} \min\{x_i, x_j\},
\end{aligned}
$$

where we have used that $|E| = n - 1$ for any tree. $\qquad\square$

# B  Proof of Main Lemmas

## B.1  Proof of Lemma 1

Recall that the event $E$ is defined as $E = (\cap_{i=1}^{K} A_i) \cap B$. First we prove that $\mathbb{P}(B^c)$ is small. Observe that if the optimal arm $\star$ is eliminated by arm $i$ at time $t$, then before time $t$ both arms are pulled the same number of times $\tau$. For any fixed realization of $\tau$, this occurs with probability at most

$$\mathbb{P}\left(\mathcal{N}(-\Delta_i, 2\tau^{-1}) \geq \sqrt{\frac{\gamma \log(TK)}{\tau}}\right) \leq \mathbb{P}\left(\mathcal{N}(0, 2\tau^{-1}) \geq \sqrt{\frac{\gamma \log(TK)}{\tau}}\right) \leq \frac{1}{(TK)^3}.$$

As a result, by the union bound,

$$\mathbb{P}(B^c) \leq \sum_{i=1}^{K} \sum_{t=1}^{T} \sum_{1 \leq \tau \leq T} \mathbb{P}\left(\text{arm } \star \text{ is eliminated by arm } i \text{ at time } t \text{ with } \tau \text{ pulls}\right) \leq \frac{1}{TK}. \quad (12)$$

Next we upper bound $\mathbb{P}(B \cap A_i^c)$ for any $i \in [K]$. Note that the event $B \cap A_i^c$ implies that the optimal arm $\star$ does not eliminate arm $i$ at time $t_{m_i} \in \mathcal{T}$, where both arms have been pulled $\tau \geq \tau_i^\star$ times. By the definition of $\tau_i^\star$, this implies that

$$\Delta_i \geq 2\sqrt{\frac{\gamma \log(TK)}{\tau}}.$$

Hence, for any fixed realizations $t_{m_i}$ and $\tau$, this event occurs with probability at most

$$\mathbb{P}\left(\mathcal{N}(\Delta_i, 2\tau^{-1}) \leq \sqrt{\frac{\gamma \log(TK)}{\tau}}\right) \leq \mathbb{P}\left(\mathcal{N}(0, 2\tau^{-1}) \leq -\sqrt{\frac{\gamma \log(TK)}{\tau}}\right) \leq \frac{1}{(TK)^3}.$$

Therefore, by a union bound,

$$\mathbb{P}(B \cap A_i^c) \leq \sum_{t_{m_i} \in \mathcal{T}} \sum_{1 \leq \tau \leq T} \mathbb{P}(\text{arm } \star \text{ does not eliminate arm } i \text{ at time } t_{m_i} \in \mathcal{T} \text{ with } \tau \text{ pulls})$$

$$\leq \frac{1}{TK^2}. \quad (13)$$

Combining (12) and (13), we conclude that

$$\mathbb{P}(E^c) \leq \mathbb{P}(B^c) + \sum_{i=1}^{K} \mathbb{P}(B \cap A_i^c) \leq \frac{2}{TK}.$$

## B.2 Deferred proof of Theorem 4

The regret analysis of the policy $\pi_{\text{BaSE}}^2$ under the geometric grid is analogous to Section 2.2. Partition the arms $\mathcal{I} = \mathcal{I}_0 \cup \mathcal{I}_1 \cup \mathcal{I}_2$ as before, and let $\Delta = \min\{\Delta_i : i \in [K], \Delta_i > 0\}$ be the smallest gap. We treat $\mathcal{I}_0, \mathcal{I}_1$ and $\mathcal{I}_2$ separately.

1. The total regret incurred by arms in $\mathcal{I}_0$ is at most

$$b \cdot \max_{i \in [K]} \Delta_i = O(b\sqrt{K}) = O\left(\frac{bK}{\Delta}\right). \quad (14)$$

2. The total regret incurred by pulling an arm $i \in \mathcal{I}_1$ is at most

$$\Delta_i \cdot \frac{t'_{m_i}}{K - \sigma_i} \leq \frac{1}{\Delta} \cdot \frac{t'_{m_i} \Delta_i^2}{K - \sigma_i} \leq \frac{2b}{\Delta} \cdot \frac{t'_{m_i-1} \Delta_i^2}{K - \sigma_i} \leq \frac{2b}{\Delta} \cdot \frac{4\gamma K \log(KT)}{K - \sigma_i},$$

where for the last inequality we have used the definition of $m_i$. Using a similar argument for $(\sigma_i : i \in \mathcal{I}_1)$ as in Section 2.2, the total regret incurred by pulling arms in $\mathcal{I}_2$ is at most

$$\sum_{i \in \mathcal{I}_1} \frac{2b}{\Delta} \cdot \frac{4\gamma K \log(TK)}{K - \sigma_i} \leq \frac{8\gamma bK \log K \log(KT)}{\Delta}. \quad (15)$$

3. The total regret incurred by pulling an arm $i \in \mathcal{I}_2$ (which is pulled $T_i$ times) is at most

$$\Delta_i T_i \leq \frac{\Delta_i^2 T_i}{\Delta} \leq \frac{4\gamma K \log(TK)}{\Delta} \cdot \frac{T_i}{t'_{M-1}} \leq \frac{8\gamma bK \log(TK)}{\Delta} \cdot \frac{T_i}{T},$$

and thus the total regret by pulling arms in $\mathcal{I}_2$ is at most

$$\sum_{i \in \mathcal{I}_2} \frac{8\gamma bK \log(TK)}{\Delta} \cdot \frac{T_i}{T} \leq \frac{8\gamma bK \log(TK)}{\Delta}. \quad (16)$$

Now combining (14) to (16) together with the inequality (6) and the choice of $b$ in (3), we arrive at the desired upper bound (5).

## B.3 Proof of Lemma 3

It is easy to show that the minimizer of $\frac{1}{n}\sum_{i=1}^{n} Q_i(\Psi \neq i)$ is $\Psi^\star(\omega) = \arg\max_{i\in[n]} Q_i(d\omega)$, and thus

$$\frac{1}{n}\sum_{i=1}^{n} Q_i(\Psi \neq i) \geq 1 - \frac{1}{n}\int \max\{dQ_1, dQ_2, \cdots, dQ_n\} = \frac{1}{n}\int\left[\sum_{i=1}^{n} dQ_i - \max_{i\in[n]} dQ_i\right].$$

By Lemmas 6 and 7, we further have

$$\begin{aligned}
\frac{1}{n}\sum_{i=1}^{n} Q_i(\Psi \neq i) &\geq \sum_{(i,j)\in E} \frac{1}{n}\int \min\{dQ_i, dQ_j\} \\
&= \sum_{(i,j)\in E} \frac{1 - \mathsf{TV}(Q_i, Q_j)}{n} \\
&\geq \sum_{(i,j)\in E} \frac{1}{2n}\exp(-D_{\mathrm{KL}}(Q_i\|Q_j)),
\end{aligned}$$

as claimed.

## B.4 Proof of Lemma 4

The proof of Lemma 4 relies on the reduction of the minimax lower bound to multiple hypothesis testing. Without loss of generality we assume that $j \in [M-1]$; the case where $j = M$ is analogous. For any $k \in [K-1]$, consider the following family $\mathcal{P}_{j,k} = (Q_{j,k,\ell})_{\ell\in[K]}$ of reward distributions: define $Q_{j,k,k} = P_{j,k}$, and for $\ell \neq k$, let $Q_{j,k,\ell}$ be the modification of $P_{j,k}$ where the quantity $3\Delta_j$ is added to the mean of the $\ell$-th component of $P_{j,k}$. We have the following observations:

1. For each $\ell \in [K]$, arm $\ell$ is the optimal arm under reward distribution $Q_{j,k,\ell}$;
2. For each $\ell \in [K]$, pulling an arm $\ell' \neq \ell$ incurs a regret at least $\Delta_j$ under reward distribution $Q_{j,k,\ell}$;
3. For each $\ell \neq k$, the distributions $Q_{j,k,\ell}$ and $Q_{j,k,k}$ only differ in the $\ell$-th component.

By the first two observations, similar arguments in (9) yield to

$$\sup_{\{\mu^{(i)}\}_{i=1}^{K}:\Delta_i\leq\sqrt{K}} \mathbb{E}[R_T(\pi)] \geq \Delta_j \sum_{t=1}^{T} \frac{1}{K}\sum_{\ell=1}^{K} Q_{j,k,\ell}^t(\pi_t \neq \ell),$$

where $Q_{j,k,\ell}^t$ denotes the distribution of observations available at time $t$ under reward distribution $Q_{j,k,\ell}$, and $\pi_t$ denotes the policy at time $t$. We lower bound the above quantity as

$$\begin{aligned}
\sup_{\{\mu^{(i)}\}_{i=1}^{K}:\Delta_i\leq\sqrt{K}} \mathbb{E}[R_T(\pi)] &\overset{(a)}{\geq} \Delta_j \sum_{t=1}^{T} \frac{1}{K}\sum_{\ell\neq k}\int \min\{dQ_{j,k,k}^t, dQ_{j,k,\ell}^t\} \\
&\geq \Delta_j \sum_{t=1}^{T_j} \frac{1}{K}\sum_{\ell\neq k}\int \min\{dQ_{j,k,k}^t, dQ_{j,k,\ell}^t\} \\
&\overset{(b)}{\geq} \Delta_j T_j \cdot \frac{1}{K}\sum_{\ell\neq k}\int \min\{dQ_{j,k,k}^{T_j}, dQ_{j,k,\ell}^{T_j}\} \\
&\geq \Delta_j T_j \cdot \frac{1}{K}\sum_{\ell\neq k}\int_{A_j} \min\{dQ_{j,k,k}^{T_j}, dQ_{j,k,\ell}^{T_j}\} \\
&\overset{(c)}{=} \Delta_j T_j \cdot \frac{1}{K}\sum_{\ell\neq k}\int_{A_j} \min\{dQ_{j,k,k}^{T_{j-1}}, dQ_{j,k,\ell}^{T_{j-1}}\}, \quad\quad (17)
\end{aligned}$$

where (a) follows by the proof of Lemma 3 and considering a star graph on $[K]$ with center $k$, and (b) is due to the identity $\int \min\{dP, dQ\} = 1 - \mathsf{TV}(P, Q)$ and the data processing inequality of the

total variation distance, and for step (c) we note that when $A_j = \{t_{j-1} < T_{j-1}, t_j \geq T_j\}$ holds, the observations seen by the policy at time $T_j$ are the same as those seen at time $T_{j-1}$. To lower bound the final quantity, we further have

$$
\begin{aligned}
\int_{A_j} \min\{dQ_{j,k,k}^{T_{j-1}}, dQ_{j,k,\ell}^{T_{j-1}}\} &= \int_{A_j} \frac{dQ_{j,k,k}^{T_{j-1}} + dQ_{j,k,\ell}^{T_{j-1}} - |dQ_{j,k,k}^{T_{j-1}} - dQ_{j,k,\ell}^{T_{j-1}}|}{2} \\
&= \frac{Q_{j,k,k}^{T_{j-1}}(A_j) + Q_{j,k,\ell}^{T_{j-1}}(A_j)}{2} - \frac{1}{2}\int_{A_j} |dQ_{j,k,k}^{T_{j-1}} - dQ_{j,k,\ell}^{T_{j-1}}| \\
&\overset{(d)}{\geq} \left(Q_{j,k,k}^{T_{j-1}}(A_j) - \frac{1}{2}\mathsf{TV}(Q_{j,k,k}^{T_{j-1}}, Q_{j,k,\ell}^{T_{j-1}})\right) - \mathsf{TV}(Q_{j,k,k}^{T_{j-1}}, Q_{j,k,\ell}^{T_{j-1}}) \\
&\overset{(e)}{=} P_{j,k}(A_j) - \frac{3}{2}\mathsf{TV}(Q_{j,k,k}^{T_{j-1}}, Q_{j,k,\ell}^{T_{j-1}}),
\end{aligned}
\tag{18}
$$

where (d) follows from $|P(A) - Q(A)| \leq \mathsf{TV}(P, Q)$, and in (e) we have used the fact that the event $A_j$ can be determined by the observations up to time $T_{j-1}$ (and possibly some external randomness). Also note that

$$
\begin{aligned}
\frac{1}{K}\sum_{\ell \neq k} \mathsf{TV}(Q_{j,k,k}^{T_{j-1}}, Q_{j,k,\ell}^{T_{j-1}}) &\leq \frac{1}{K}\sum_{\ell \neq k} \sqrt{1 - \exp(-D_{\mathrm{KL}}(Q_{j,k,k}^{T_{j-1}} \| Q_{j,k,\ell}^{T_{j-1}}))} \\
&= \frac{1}{K}\sum_{\ell \neq k} \sqrt{1 - \exp\left(-\frac{9\Delta_j^2 \mathbb{E}_{P_{j,k}}[\tau_\ell]}{2}\right)} \\
&\leq \frac{K-1}{K} \sqrt{1 - \exp\left(-\frac{9\Delta_j^2}{2(K-1)}\sum_{\ell \neq k} \mathbb{E}_{P_{j,k}}[\tau_\ell]\right)} \\
&\leq \frac{K-1}{K} \sqrt{1 - \exp\left(-\frac{9\Delta_j^2 T_{j-1}}{2(K-1)}\right)} \leq \frac{3}{\sqrt{K}} \cdot \sqrt{\Delta_j^2 T_{j-1}} \leq \frac{1}{12M},
\end{aligned}
\tag{19}
$$

where the first inequality is due to Lemma 6, the second equality evaluates the KL divergence with $\tau_\ell$ being the number of pulls of arm $\ell$ before time $T_{j-1}$, the third inequality is due to the concavity of $x \mapsto \sqrt{1 - e^{-x}}$ for $x \geq 0$, the fourth inequality follows from $\sum_{\ell \neq k} \tau_\ell \leq T_{j-1}$ almost surely, and the remaining steps follow from (11) and simple algebra.

Combining (17), (18) and (19), we conclude that

$$
\sup_{\{\mu^{(i)}\}_{i=1}^K : \Delta_i \leq \sqrt{K}} \mathbb{E}[R_T(\pi)] \geq \Delta_j T_j \left(\frac{P_{j,k}(A)}{2} - \frac{1}{8M}\right) \geq \sqrt{K}T^{\frac{1}{2-2^{1-M}}} \cdot \frac{1}{72M}\left(\frac{P_{j,k}(A)}{2} - \frac{1}{8M}\right).
$$

Note that the previous inequality holds for any $k \in [K-1]$, averaging over $k \in [K-1]$ yields

$$
\begin{aligned}
\sup_{\{\mu^{(i)}\}_{i=1}^K : \Delta_i \leq \sqrt{K}} \mathbb{E}[R_T(\pi)] &\geq \sqrt{K}T^{\frac{1}{2-2^{1-M}}} \cdot \frac{1}{72M}\left(\frac{1}{2(K-1)}\sum_{k=1}^{K-1} P_{j,k}(A) - \frac{1}{8M}\right) \\
&\geq \frac{1}{576M^2} \cdot \sqrt{K}T^{\frac{1}{2-2^{1-M}}},
\end{aligned}
$$

where in the last step we have used that $p_j \geq \frac{1}{2M}$. Hence, the proof of Lemma 4 is completed.

## B.5   Proof of Lemma 5

Recall that the event $A_j$ can be determined by the observations up to time $T_{j-1}$ (and possibly some external randomness), the data-processing inequality gives

$$
|P_M(A_j) - P_{j,k}(A_j)| \leq \mathsf{TV}(P_M^{T_{j-1}}, P_{j,k}^{T_{j-1}}).
$$

Note that each $P_{j,k}$ only differs from $P_M$ in the $k$-th component with mean difference $\Delta_j + \Delta_M$, the same arguments in (19) yield

$$\frac{1}{K-1}\sum_{k=1}^{K-1}\mathsf{TV}(P_M^{T_{j-1}}, P_{j,k}^{T_{j-1}}) \leq \frac{1}{K-1}\sum_{k=1}^{K-1}\sqrt{1 - \exp(-D_{\mathsf{KL}}(P_M^{T_{j-1}}\|P_{j,k}^{T_{j-1}}))}$$

$$= \frac{1}{K-1}\sum_{k=1}^{K-1}\sqrt{1 - \exp\left(-\frac{(\Delta_j + \Delta_M)^2}{2}\mathbb{E}_{P_M}[\tau_k]\right)}$$

$$\leq \sqrt{1 - \exp\left(-\frac{2\Delta_j^2}{K-1}\mathbb{E}_{P_M}\left[\sum_{k=1}^{K-1}\tau_k\right]\right)}$$

$$\leq \sqrt{1 - \exp\left(-\frac{2\Delta_j^2 T_{j-1}}{K-1}\right)} \leq \frac{1}{2M},$$

where we define $\tau_k$ to be the number of pulls of arm $k$ before the time $T_{j-1}$, and $\sum_{k=1}^{K-1}\tau_k \leq T_{j-1}$ holds almost surely. The previous two inequalities imply that

$$|P_M(A_j) - p_j| \leq \frac{1}{K-1}\sum_{k=1}^{K-1}|P_M(A_j) - P_{j,k}(A_j)| \leq \frac{1}{2M},$$

and consequently

$$\sum_{j=1}^{M}p_j \geq P_M(A_M) + \sum_{j=1}^{M-1}\left(P_M(A_j) - \frac{1}{2M}\right) \geq \sum_{j=1}^{M}P_M(A_j) - \frac{1}{2}. \qquad (20)$$

Finally note that $\cup_{j=1}^{M}A_j$ is the entire probability space, we have $\sum_{j=1}^{M}P_M(A_j) \geq P_M(\cup_{j=1}^{M}A_j) = 1$, and therefore (20) yields the desired inequality.