[Reviews · NeurIPS 2019]

Reviewer 1



i. Novelty and significance: The problem addressed in the paper seems new and the proposed "BaSE" algorithm seems novel for the purpose. The technical contribution of the paper seems sound as authors analysed both minimax and adaptive regret of their proposed algorithm with pre-specified (hence fixed), and (more natural and challenging) adaptive batch sizes, and also prove a matching lower bound guarantee, establishing optimality of their proposed method. ii. Clarity on some results: a) Its confusing that as opposed to what is claimed in Cor 1, I do not see how Thm. 1 recovers the optimal regret O(\srt{KT} bound of classical MAB framework (i.e. when M=T), unless of course T \to infty which is an asymptotic guarantee. --- A thorough derivation of Cor. 1 statement would be appreciated. b)I am also surprised with the matching lower bound statement (Thm. 2), as for M=T it definitely seems to be higher than the classical \Omega(\sqrt{KT}) lower bound for MABs -- what am I missing? c) Intuitively, the learner is supposed to have better control for the data-driven grid setting --- It is however reflected from the analysis that the regret bounds obtained for the two setting are exactly similar (Thm. 1 and 4 or Thm 2 and 3). Why is that -- an elaboration of Line 154-157 would be useful. iii. Experiments: Its however slightly disappointing that it does not provide any empirical studies to validate the theoretical guarantees. iv. Organization and presentation: The paper is overall well written and easy to follow.

Reviewer 2



In the multi-armed bandits problem, it is assumed that the player interacts with the environment in a round-robin manner: the decision in round t can depend on the feedback received during rounds 1,...,t-1. In some applications, this is not realistic, e.g., in clinical trials, you may test a drug on several patients simultaneously, and receive the feedback all at once. This paper studies such batched multi-armed bandits problems, where the player can have M rounds of interaction with the environment. The rounds of interaction can be predetermined by the policy (the static grid model), or it can be based on the received data (the data-driven grid). The case M=T corresponds to the classical multi-armed bandits problem. The paper provides an algorithm (called BaSE) for the static grid model, which is based on eliminating bad arms. They give minimax and problem-dependent regret upper bounds for this algorithm (Theorem 1). Certainly, these also give upper bounds for the data-driven model, since in the data-driven model the player has more power and so the regret cannot be larger. Theorem 1 implies that M=O(log log T) batches are sufficient to obtain optimal minimax regret, and M = O(log T) batches are sufficient to obtain optimal problem-dependent regret. The paper also discusses lower bounds. In Theorem 2, information-theoretic lower bounds are proved for the static model, which are within polylog factors of the upper bounds. Finally, in Theorem 3, information-theoretic lower bounds are proved for the data-driven model, which are within M^2 times polylog factors of the upper bounds. The paper is well-written, the technical contributions are new and solid, and the results are strong (especially the static model). A systematic treatment of this problem is missing in the literature, and this paper fills this gap. It is likely that some of the technical lemmas of this paper (e.g., Lemma 3) may be useful for other researchers. I haven't checked the proofs. Comments for authors: - using the word "adaptive" in definition (2) is unfortunate. Adaptive usually means that the algorithm can adapt its future decisions based on the past. So, when the reader first sees the word adaptive, she thinks you're talking about choosing the interaction rounds t1, t2,... So I suggest using, for instance, minimax bounds vs. problem-dependent bound. You can then use "adaptive grid" instead of "data-driven grid." - An interesting open question is finding the correct rates for the data-driven grid model. Maybe pose this explicitly as an open question. - The algorithm pseudocode is not very clear. Instead of line (a), write something like: during rounds t_{m-1} to t_m - 1, pull each arm of A for the same number of times. - Fix author names in the reference [BPR13] == After reading the authors responses: thanks. Please add these responses to the revised version as well.

Reviewer 3



- originality The task is a straightforward extension of the existing work [PRCS16]. However, the algorithm is not a simple extension of the existing work. - quality, clarity Their writing is clear and easy to follow. In their analysis, no errors can be found. - significance I believe this is a significant work as the batched setting is common in practice, and this work gives a complete characterization on the minimax regret on the batched multi-armed bandits.

[Author Response · NeurIPS 2019]

We are grateful to the reviewers for the comments. Below we respond to each and every point of the reviewers (where
we refer to the same set of references in the submission):

**Reviewer 1**: We thank the reviewer for the positive comments. Our corollaries follow from the fact that $T^{\frac{1}{2-2^{1-M}}} =$
$T^{\frac{1}{2-2^{1-\log_2 \log T}}} = \sqrt{T} \cdot T^{\frac{1}{2(\log T - 1)}} = \sqrt{T} \cdot e^{\frac{\log T}{2 \log T - 1}} = \Theta(\sqrt{T})$ for the minimax regret when $M = \log_2 \log T$ and
$T^{\frac{1}{M}} = T^{\frac{1}{\log T}} = e = \Theta(1)$ for the adaptive regret when $M = \log T$. Hence, under the conditions of Corollary 1, the
optimal regrets $\Theta(\sqrt{KT})$ or $\Theta(K \log T)$ are attained within logarithmic factors. The lower bound parts are also similar.
Moreover, although the data-driven grid is *no weaker* than the static grid by definition, our result shows that it is also *no*
*stronger* than the static grid for the batched bandit problem in the sense that a static grid suffices to essentially achieve
the optimal regrets. Hence, one of our contributions is to show that data-driven grids do not improve much over the
static ones, which is the focus of Theorem 3. In the final version we will make these points clearer and more explicit.

We appreciate the reviewer's great suggestions of adding experiments. We have done the following: first, we numerically
investigate the regret dependence on parameters $(T, K, M, \Delta)$ and the choices of different grids. For example, Figure
(a) plots the average regrets of BaSE under different grids as a function of $M$, as well as the regrets of the centralized
algorithm UCB1 without batch constraints [ACBF02]. Here we take $T = 5 \times 10^4$, $K = 3$, $\gamma = 1$, standard normal arms
with mean $(0.6, 0.5, 0.5)$, and arithmetic grid stands for the grid with equal spacing. We observe from Figure (a) that a
very small number of batches (e.g., $M = 4$) are sufficient for BaSE to roughly achieve the centralized performance
under the minimax grid. Second, since policies for batched bandits with $K > 2$ arms are missing before our work, we
only compare our BaSE policies with the ETC policies in [PRCS16] for two-armed bandits. Under the same setting
above, Figure (b) shows that BaSE achieves lower regrets than ETC. Complete experiments will be in the final version.

(a) Comparison of grids used in BaSE.          (b) Comparison of BaSE and ETC.

**Reviewer 2**: We thank the reviewer for the positive comments. As suggested, in the final version we will change
"adaptive regret" into the "problem-dependent regret", "adaptive grid" into "geometric grid", and "data-driven grid" into
"adaptive grid". We will also fix the error in the reference [BPR13] and improve the algorithm pseudocode.

The correct rates for the data-driven grid model are great questions. We believe that the additional $M^{-2}$ factor in
Theorem 3 is an artifact in our proof, and we conjecture that the same lower bounds in Theorem 2 still hold. We would
also like to emphasize that the $M^2$ multiplicative gap is at most poly-logarithmic in $T$ in our batched bandit problem,
for we may apply known centralized lower bounds (e.g., $\Omega(\sqrt{KT})$ for the minimax regret in the fully online case)
whenever $M \gg \log T$. However, we believe that resolving this question will be helpful for other problems with limited
rounds of adaptivity, and in the final version we will explicitly state it as an open problem.

**Reviewer 3**: We thank the reviewer for the positive comments. The adversarial setting with batch constraints is a great
question, where we believe that proper definitions of the adversary and adversarial regret would be important. It is not
hard to show that when the adversary can choose the rewards in $[0, 1]$ arbitrarily and the cumulative reward is compared
with the optimal fixed arm in hindsight, the regret will be at least $\Omega(T/M)$ if $2M \le K$. Consequently, for small $M$ the
regret is $\Omega(T)$, as opposed to our current results in the stochastic setting. Hence, to obtain illuminating results we need
to restrict the power of the adversary or change the definition of the regret, which would be interesting future directions.

The analysis of any pre-specified grids is also a great question. We conjecture that our BaSE algorithm is nearly optimal
for any fixed grids, and the minimax regret is $\tilde{\Theta}(\sum_{i=1}^{M}(t_i - t_{i-1})\sqrt{K/t_{i-1}})$. Intuitively, in the $i$-th batch $(t_{i-1}, t_i]$,
all remaining arms in the BaSE algorithm have been pulled at least $t_{i-1}/K$ times and thus have suboptimality gap
$\tilde{O}(\sqrt{K/t_{i-1}})$ with high probability, then the upper bound follows. The lower bound can be established using Lemma
2, which may be off an $M^{-1}$ factor after choosing $\Delta$ carefully. We will also fix the error in the reference [BPR13].

[Meta-Review · NeurIPS 2019]

The paper contributes significant results that quantify the impact, and optimal use, of partial information due to sub-sampling in stochastic multiarmed bandits -- an important class of online learning problems. In a sense this is an extension of partial information in the "space" domain (bandit arm feedback) to the "time" domain, where it is not possible to collect a sample of feedback in every round. The submission was unanimously appreciated by all reviewers and this was also reflected in the post-response discussion that ensued among the reviewers. [This meta-review was reviewed and revised by the Program Chairs]